# Influence of surprise on reinforcement learning in younger and older adults

Christoph Koch[1,2], Ondrej Zika [1,3], Rasmus Bruckner [1,4], Nicolas W. Schuck [1,2,3]*

**1** Max Planck Institute for Human Development, Berlin, Germany, **2** Institute of Psychology, Universität Hamburg, Hamburg, Germany, **3** Max Planck UCL Centre for Computational Psychiatry and Aging Research, Berlin, Germany, and London, United Kingdom, **4** Department of Education and Psychology, Freie Universität Berlin, Berlin, Germany

* nicolas.schuck@uni-hamburg.de

**Data Availability Statement:** All code related to this work has been made openly available at https://github.com/koch-means-cook/pedlr. The data collected in the experiment is published as a datalad repository (https://www.datalad.org) at

## Abstract

Surprise is a key component of many learning experiences, and yet its precise computational role, and how it changes with age, remain debated. One major challenge is that surprise often occurs jointly with other variables, such as uncertainty and outcome probability. To assess how humans learn from surprising events, and whether aging affects this process, we studied choices while participants learned from bandits with either Gaussian or bimodal outcome distributions, which decoupled outcome probability, uncertainty, and surprise. A total of 102 participants (51 older, aged 50–73; 51 younger, 19–30 years) chose between three bandits, one of which had a bimodal outcome distribution. Behavioral analyses showed that both age-groups learned the average of the bimodal bandit less well. A trial-by-trial analysis indicated that participants performed choice reversals immediately following large absolute prediction errors, consistent with heightened sensitivity to surprise. This effect was stronger in older adults. Computational models indicated that learning rates in younger as well as older adults were influenced by surprise, rather than uncertainty, but also suggested large interindividual variability in the process underlying learning in our task. Our work bridges between behavioral economics research that has focused on how outcomes with low probability affect choice in older adults, and reinforcement learning work that has investigated age differences in the effects of uncertainty and suggests that older adults overly adapt to surprising events, even when accounting for probability and uncertainty effects.

## Author summary

Learning is a skill that requires a finely adjusted process of extracting just the right information from past experiences to benefit future choices. As we age, this process begins to alter, changing how we react to ambiguity, risk or uncertainty. One challenging aspect of learning is that sometimes we will encounter very surprising consequences of our actions, raising the question whether we should assign more or less weight to these events. We know relatively little about how humans react to these surprises and how age affects learning from surprising outcomes. To learn more about this question, we asked 51 older and

https://gin.g-node.org/koch_means_cook/pedlr-main-data.git with the DOI: 10.12751/g-node.9gm3lt. All derivatives of the collected data supporting the findings of this study, for instance data after model fitting and rendered code books are openly available at https://gin.g-node.org/koch_means_cook/pedlr-derivatives.git with the DOI: 10.12751/g-node.tsq6sg.

**Funding:** o NWS was funded by the Federal Government of Germany and the State of Hamburg as part of the Excellence Initiative, a Starting Grant from the European Union (ERC- StG-REPLAY-852669). NWS and OZ were supported by an Independent Max Planck Research Group grant awarded to NWS (M.TN.A.BILD0004). During the work on his dissertation, CK was a pre-doctoral fellow of the International Max Planck Research School on the Life Course (LIFE; participating institutions: Max Planck Institute for Human Development, Freie Universität Berlin, Humboldt-Universität zu Berlin, University of Michigan, University of Virginia, University of Zurich). RB was supported by DFG (Deutsche Forschungsgemeinschaft) grant 412917403. The funders had no role in study design, data collection and analysis, decision to publish, or preparation of the manuscript. While working on this project, NWS's salary was paid partly by the Max-Planck-Gesellschaft and partly by the ERC.

**Competing interests:** The authors have declared that no competing interests exist.

51 younger adults to play a reinforcement learning task that confronted them with surprising outcomes and analyzed their choices. We found that both age groups showed heightened sensitivity to surprising outcomes that resulted in distinctive behavioral adjustments. Notably, older adults weighted these surprising events more than younger adults. Comparing the choices of participants with computational models that incorporated surprises in different ways, we found a model that modulated its learning with the amount of surprise to mimic participants' choices best. The results help to better understand the role of past surprising events during learning in older and younger adults.

## Introduction

Aging changes how humans learn and decide in ways that can affect important life choices, such as monetary or health decisions [1–4]. Older adults for instance differ in how their decision making incorporates risk [5, 6], ambiguity [2], uncertainty [3], feedback [7–9], or explicit memory [10, 11]. Two common factors that play a role in all of these findings are how rare events are, and how much they differ from events encountered otherwise.

How humans react to rare events has been the focus of much interest. Seminal work on prospect theory [12, 13] has for instance shown that college-aged adults overweight events with low probabilities during decision making, and perceive relatively less gains with larger outcomes, which could explain why people are often uncertainty-averse in the gain domain [14]. Differential processing of rare outcomes can also influence decision making informed by past events, since memory tends to be better for values at the edges of distributions [15–17]. Memory for events associated with less expected outcomes (which are often rare too) also appears to be better [18, 19].

Past aging research has studied related but not identical aspects of decision making [3, 5, 20]. Nassar et al. [3], for instance, investigated learning of older and younger adults in changing environments characterized by so called non-stationary bandits, i.e. a scenario in which the rewards associated with different actions change over time. They specifically focused on how participants modulated their learning rates in response to outcome deviations that reflected a true shift of the bandit mean (due to an environmental change point) versus merely a random deviation due to variability around each bandit's mean, which represent a mix of outcome probability and deviation from previous events. Nassar et al. suggested that in this setup uncertainty processing, but not surprise processing, is impaired in older relative to younger adults [3]. These effects might arise from a simplified learning strategy that reduces cognitive resource expenditure, making older adults less sensitive to smaller prediction errors that can be attributed to uncertainty compared to larger and more surprising prediction errors [21]. However, this line of work leaves open the question of how surprise affects learning in older adults when the surprising event does not signal a fundamental change point and, therefore, dictates a lower learning rate [22].

Other work in the domain of decisions from description suggests that older adults overweight low probability events in the gain domain (i.e. show more risk-seeking behavior), compared to younger adults [20]. This work has focused purely on how the stated probability of events affects decision making. In contrast, when decisions are based on learned probabilities, referred to as decisions from experience [23, 24], age-related differences in choice behavior and risk-taking often differ compared to decisions from description, where age differences in risk preferences depend on the exact choice scenario [5].

Therefore, we aimed to examine age-related learning and decision making differences in an experience-based choice task with stationary outcome probabilities. We specifically studied the effects of outcomes that are highly *surprising*, i.e. differ significantly from most previous outcomes. We stipulated that surprise could affect the learning rate with which participants update their expectations in a trial and error setting, even when dissociated from the effects of probability. In line with previous work, we expected that surprise would have a greater effect on older adults, as compared to younger adults. Taking a reinforcement learning (RL) perspective [25, 26], we conceptualized surprise as the absolute prediction error (PE), i.e. the deviation of an observed outcome from the current expectation. While standard RL theory assumes that prediction errors are weighted by a constant learning rate parameter $\alpha \in [0, 1]$, we hypothesized that learning rates are modulated by the absolute PE, i.e. surprise of a given trial. Our idea specifically predicts that surprise impacts learning *immediately*, i.e. affects the update on the very same trial that caused the surprise. In turn, this is akin to a process that gives more weight to an event not based on its probability, but on its associated surprise which dissociates our proposal from previous work where learning rates only ramp up future, but not current, learning [27], and prediction error magnitude is often confounded with outcome probability [27–31].

We designed a novel task in which participants learned from outcomes drawn from a stationary bimodal distribution (a non-changing distribution with two peaks) that yielded a number of benefits when studying the effects of surprise on learning. First, compared to changing environments [3], stationary bandits reflect a much simpler case that arguably occurs quite often in everyday life, and allowed us to work with much simpler formulations of surprise and uncertainty (see below). Second, a bimodal distribution has a second peak of outcomes that are far from the mean, but still relatively probable, which makes it possible to decouple an event's probability from its surprise. In unimodal Gaussian distributions, prediction errors, outcome probabilities, and magnitude are correlated. However, this correlation is lessened or absent in long-tailed or bimodal distributions, where outcomes with a relatively small difference from the mean can have a probability as low as outcomes much further from the mean. Using this setup also makes it possible to simultaneously differentiate surprise from uncertainty. We defined uncertainty according to Platt & Huettel [14] as the absence of knowledge about which choice will produce which outcome. To differentiate surprise from uncertainty, we hence contrasted the trialwise fluctuating absolute PE with the trailing average of surprise, which reflects how much uncertainty participants experienced in the past [3, 27, 32]. Finally, using our task we could also investigate whether any form of asymmetric learning from positive versus negative prediction errors [33, 34], which often has been observed in an aging context [7, 35], played a role in explaining age differences in learning.

## Materials and methods

### Ethics statement

All participants provided informed consent and the study was approved by the ethics committee of the Max Planck Institute for Human Development (approval number: N-2020-01).

### Participants

Participants were recruited using *Prolific* (www.prolific.co). We collected data of 64 younger (18–30 years, mean age 24.42) and 56 older adults (50–73 years, mean age 57.18). Eighteen participants (13 younger, 5 older) were excluded from all analyses due to insufficient task performance across both runs; 17 participants (12 younger, 5 older) did not show significant above-chance performance in the easiest task condition (using a binomial test against chance in low

vs. high bandit trials, see below), and one young adult had a disproportionate amount of errors in guided choice trials compared to the rest of the sample (more than three SDs from mean of distribution). The effective sample of choice trials therefore consisted of 102 participants (51 younger, 51 older).

To ensure high data quality in the analyses of the estimation trials, of which only 16 existed per run (see below), we applied additional exclusion criteria exclusively for these analyses. Specifically, data from runs in which a participant did not show any overall difference in estimates of the low versus high bandit, or did not show any variance in their estimates, were excluded. This resulted in the exclusion of 12 runs from 10 participants for indistinguishable low vs. high estimates (no sig. difference in paired t-test) and of 2 runs from one participant due to no variance in submitted answers. Estimation-based analyses therefore included data of 99 participants (49 younger, 50 older), out of which 8 participants had only one remaining run.

The experiment lasted about 60 minutes, participants were remunerated with a baseline payment of 7.5 GBP plus a performance based bonus of up to 3 GBP (see below).

## Value-based learning task

The task consisted of two runs of a value-based choice task. In each run participants learned about three different bandits that provided rewards drawn from distributions with a low, medium or high mean (outcome range 1–100 points, details see below). We will refer to these bandits as the *low*, *mid*, and *high* bandit, respectively (see below for details, and Fig 1B). Each bandit was indicated by a different Japanese Hiragana symbol (randomly assigned across participants). Participants had to learn about each bandits' value through trial and error and did

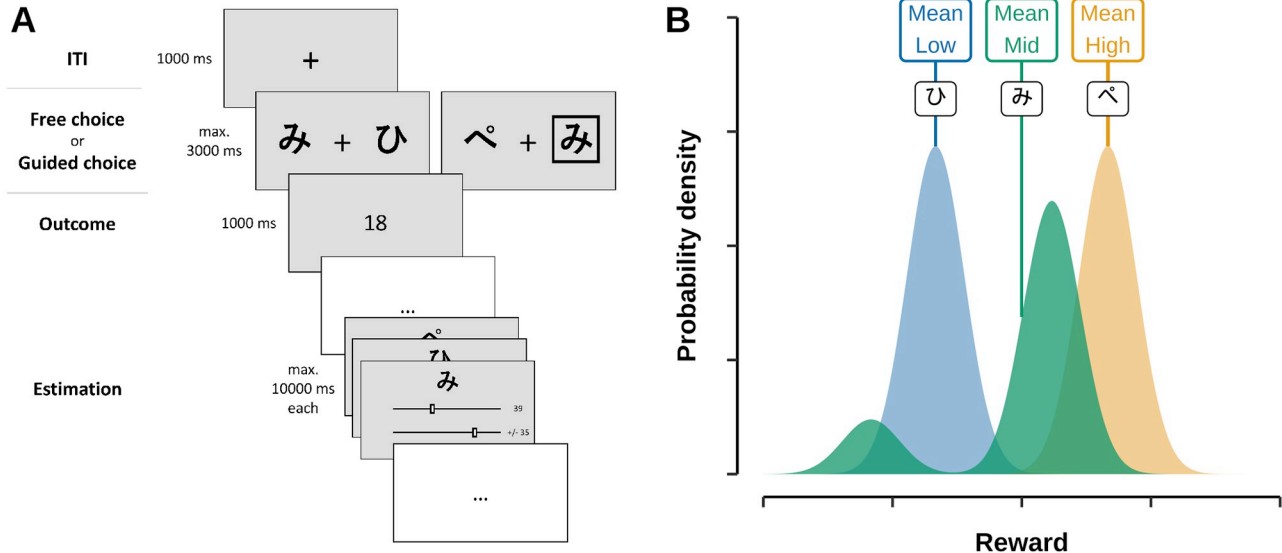

**Fig 1. Task and design. A**: Schematic of task procedure. The first three steps show the procedure of a free choice or guided choice trial. After a brief inter trial interval of 1000 ms participants were confronted with a choice between two bandits. In free choice trials participants could freely choose either of both bandits. In guided choice trials participants were instructed to choose the framed option. After a choice was made the outcome of said choice was displayed for 1000 ms. Occasionally, participants had to complete estimation trials in which they had to estimate how many points they will get when choosing each bandit as well as the range in which the points may vary. **B**: Schematic of reward distributions. Each bandit was linked to one of three reward distributions: one with a low, medium, and high average reward. Means of subsequent distributions were equidistant (16.66). While the low and high distribution were Gaussian the mid distribution was bimodal with the two modes being 35 points apart. The smaller mode was always to the left of the greater mode and made up 20% of possible rewards. The absolute means of distributions varied between runs while the distance between distributions and distance between modes of the mid distribution never changed.

not receive any information about reward distributions or reward schedules besides the obtained points. Points collected were translated into a monetary bonus of up to 3 GBP at the end of the experiment. Prior to the task all participants went through identical, text-based instructions and a short training period that conveyed information about the different trial types (see below), but not the differences in the underlying distributions.

**Reward distributions.** To answer our main question about how participants learn from surprising outcomes characterized by a large PE, we manipulated the reward distributions of the different bandits (see Fig 1B). Rewards of the low and high average bandit were drawn from regular Gaussian distributions with a standard deviation of 5.55 points. The means of both Gaussians were fixed within each run and always chosen in a way that they were 33.33 points apart. The rewards of the mid bandit followed a bimodal mixture distribution composed of two Gaussians (each sd = 5.55 points): a main mode (80% of outcomes) and a smaller mode (20% of outcomes) with a distance of 35 points. The total mean of the mid bandit was equidistant from the means of the low and high bandits, 16.66 points away. Notably, the smaller mode of the mid bandit was in fact lower than the mean of the low bandit (despite the general mean of the mid bandit being higher). This asymmetrical outcome distribution of the mid bandit was central to the idea of the task: while the medium bandit delivered higher outcomes than the low bandit *on average*, in 20% of choices it produced a low outcome that was sampled from the lower mode of the distribution. We therefore expected that over- or underweighting of surprising outcomes would specifically bias participants' decisions between and estimations of the low and medium bandit.

At the start of a the second run a separate set of three bandits was introduced. The absolute means of each distribution changed between runs by up to 14.8 points, while their relative structure (distance between means, distribution shapes) remained the same. Participants were made aware that outcomes and symbols were changed at the start of the new run, and that rewards were constrained to lie between 0 and 100 points.

**Free/guided choice trials.** On each choice trial participants had to decide between two bandits, ensuring that all pairwise bandit combinations (low-mid, mid-high, and low-high) appeared with equal frequency within each run. In free choice trials (192 of 240 trials per run), participants could freely chose between the offered bandits within a maximum of 3000 ms. After a choice, the outcome was displayed for 1000 ms, followed by a fixation cross (1000 ms) to allow for preparation for the next trial. Not responding within the maximum of 3000 ms resulted in 0 points and a hint to respond faster. To make sure all bandits were sampled regularly, the remaining trials consisted of guided choice trials (48 of 240 trials per run). In these trials, participants were instructed to choose the bandit that was marked with a frame. Choices of the unmarked bandit resulted in no points and a reminder to choose the framed option without displaying the bandit's outcome before participants moved on to the next trial. All other task aspects were kept the same and correct choices awarded points as usual. Choice trials are illustrated in Fig 1A at the top.

**Estimation trials.** Each run also included 16 estimation trials in which participants were asked to estimate how many points would be obtained from a bandit, and to which degree the outcomes may vary (Fig 1A, bottom). Estimates were collected for all three bandits and had to be provided by adjusting two independent sliders that ranged between 0 and 100 points for the average estimate and from −50 to +50 for the range estimate (with a step-size of 1, and a maximum decision time of 10 seconds). No feedback about their estimation was provided and participants could not earn points for accurate estimations. Estimation trials occurred at pseudo-random times within the run, assuring that there were no estimation trials within the first 10 choice trials, all estimation trials were at least 10 choice trials apart (on average, estimation trials were separated by 14.98 choice trials), and 4–5 estimation trials occurred immediately after

a guided choice trial of the mid bandit that produced an extreme, low outcome drawn from the smaller mode of the bimodal distribution (see below).

## Statistical analyses

Behavioral analyses were done using linear mixed effects (LME) models with fixed effects of interest, such as bandit comparison (which bandits were presented to choose from, 3 levels, low-mid, mid-high, low-high), run number (2 levels), and age group (2 levels: older vs. younger adults). Models also included a random effect (intercept) of participant. The first of these models investigated overall performance (percentage of correct free choice trials) taking the form of

$$
\begin{aligned}
\text{Performance}_{\text{free}} \quad &= \beta_0 + \gamma_{0,k} + \beta_1 \text{Group}_{\text{Age}} + \beta_2 \text{Run} + \beta_3 \text{Comp}_{\text{Bandit}} + \\
&\quad \beta_4 \text{Group}_{\text{Age}} * \text{Run} + \beta_5 \text{Group}_{\text{Age}} * \text{Comp}_{\text{Bandit}} + \\
&\quad \beta_6 \text{Run} * \text{Comp}_{\text{Bandit}} + \beta_7 \text{Group}_{\text{Age}} * \text{Run} * \text{Comp}_{\text{Bandit}} ,
\end{aligned} \tag{1}
$$

where $\beta_0$ and $\gamma_{0,k}$ denote global and subject-specific intercepts, $\beta_1$ to $\beta_3$ represent the fixed main effects of age group, run number, and bandit comparison, and $\beta_4$ to $\beta_7$ their respective interactions. A similar model was used to investigate choice speed (reaction times; all reaction times were collected in milliseconds and log-transformed before entering any analyses, equation identical to right hand side of Eq 1). To investigate the effect of large prediction errors, we analyzed free choices in low-mid trials before and after participants encountered a surprising outcome of the mid bandit's lower mode (below the distributions 20th percentile, on average n = 4.71 and n = 4.44 choices per run/participant, respectively). This was compared to choices in low-mid trials following less surprising outcomes from the 20th to 40th percentile of the distribution. The model for this analysis was specified as

$$
\begin{aligned}
p(\text{Choice}_{\text{mid}} | \text{Trial}_{\text{low-mid}}) \quad &= \beta_0 + \gamma_{0,k} + \beta_1 \text{Position} + \beta_2 \text{Group}_{\text{Age}} + \beta_3 \text{Run} + \\
&\quad \beta_4 \text{Position} * \text{Group}_{\text{Age}} + \beta_5 \text{Position} * \text{Run} + \\
&\quad \beta_6 \text{Group}_{\text{Age}} * \text{Run} + \beta_7 \text{Position} * \text{Group}_{\text{Age}} * \text{Run}
\end{aligned} \tag{2}
$$

and included a fixed effect of position relative to the large surprise, i.e. absolute PE (pre vs. post, $\beta_1$), in addition to the fixed effects of age group ($\beta_2$) and run ($\beta_3$), their interaction terms ($\beta_4$ to $\beta_7$) as well as a global and participant-specific intercept ($\beta_0$ and $\gamma_{0,k}$, respectively).

Statistical inference was done through $\chi^2$ likelihood ratio tests to determine whether the inclusion of a particular fixed effect in the model provided a significantly better fit (R package lme4 [36]). Posthoc test were done using the emmeans package for R [37] and were corrected for multiple comparisons applying Šidák correction.

**Analyses of estimation trials.** To check if estimates of each bandit deviated from their true average outcome, participant responses were compared to the running mean of collected outcomes of each bandit. Differences between a bandit's true average and participants' estimates were calculated to provide a measure of under- or over-estimation. Since learning in the initial trials will cause the estimated averages to fluctuate (in ways that could depend on participants unknown a-priori expectations of average outcomes), we considered only the second half of estimation trials for further analysis. Over- or underestimations were assessed using one-sample t-tests separately for each bandit and age group. P-values were Bonferroni-Holm corrected.

We also compared the difference in estimates of two bandits in regard to their objective running average difference. Because our main hypothesis concerned biases in the mid bandit, we focused on the estimated differences between the low and mid bandits, and mid and high bandits. Subtracting the estimated differences from the corresponding objective differences yielded a measure of distortion in perceived distance between bandits for each comparison, whereby values lower than zero represent an underestimation of distance. This measure of distortion was analyzed using a LME model specified as

$$
\begin{aligned}
\text{Distortion}_{\text{Estimate}} \quad &= \beta_0 \,+\, \gamma_{0,k} \,+\, \beta_1 \, \text{Group}_{\text{Age}} \,+\, \beta_2 \, \text{Run} \,+\, \beta_3 \, \text{Option} \,+ \\
&\quad \beta_4 \, \text{Group}_{\text{Age}} * \text{Run} \,+\, \beta_5 \, \text{Group}_{\text{Age}} * \text{Option} \,+ \quad\quad (3)\\
&\quad \beta_6 \, \text{Run} * \text{Option} \,+\, \beta_7 \, \text{Group}_{\text{Age}} * \text{Run} * \text{Option}
\end{aligned}
$$

with fixed effects of age group ($\beta_1$), run ($\beta_2$), and available options (low-mid vs. mid-high, $\beta_3$) as well as the respective interaction terms ($\beta_4$ to $\beta_7$) and a global and participant-specific intercept ($\beta_0$ and $\gamma_{0,k}$, respectively).

## Computational models

We applied four different RL models, plus two combination models, to participants' choice data, (details see below in the Results section). All models were based on a delta-rule updating mechanism that yielded a recency-weighted value estimate of each bandit, but differed with respect to the assumptions about the learning rate and the influence of uncertainty on choice.

**Rescorla-Wagner model.** As a baseline model we used a standard *Rescorla-Wagner* model (*RW* model, [38, 39]), in which the value of each bandit is the recency-weighted average of associated rewards, as described in the Results section below (Eq 8). We used a logistic regression approach for fitting the RW value predictions to participants choices. Specifically, the probability to chose bandit *k* over bandit *l*, given the model values was:

$$
p(k|V_{\cdot,t}) = \frac{1}{1 + e^{-(\beta_0 + \beta_1 V_{l,t} + \beta_2 V_{k,t})}} = \sigma\big(\beta_0 + \beta_1 V_{l,t} + \beta_2 V_{k,t}\big) \quad , \quad\quad (4)
$$

where $\sigma(\cdot)$ indicates the logistic function and the parameters $\beta_0$, $\beta_1$ and $\beta_2$ reflect the intercept and the influence of the values of bandits *k* and *l*, respectively, determined using maximum likelihood estimation, as implemented in function glm in R (R stats; [40]).

We performed some sanity checks to confirm that our baseline *RW* model worked in principle and participants' choices conformed with reinforcement learning mechanisms. As expected, the probability of choosing the right bandit was positively influenced by the value of the right bandit calculated by the *RW* mechanism in both age groups (avg. $\beta$ younger: 0.16 [CI: 0.12–0.20], t(50) = 7.99, p < .001; older: $\beta$ = 0.18 [0.15–0.22], t(50) = 10.40, p < .001), while the reverse was true for the left bandit (both t(50) < −8.96, p < .001). The betas of the right and left bandit values correlated with the average percent of correct choices in low-mid bandit comparisons at r = .26 (t(100) = 2.74, p = .015) and r = − .22 (t(100) = − 2.26, p = .026, Bonferroni-Holm corrected), for the right and left bandit, respectively. Hence, participants performed the task in a manner generally consistent with reinforcement learning models.

**Valence model.** Previous studies have presented compelling evidence for differences in the way humans learn from positive and negative feedback [41–43] such as positivity or negativity biases [44–46]. Older age has been shown to influence this difference [35, 47]. Since the form of the bimodal distribution of the mid bandit introduced large prediction errors that were predominantly negative and we investigated different age groups we wanted to quantify the influence of prediction error valence on choices. An additional model therefore resulted

from modifying Eq 8 so that the learning rate could differ depending on the valence of the prediction error, as specified in the results section (Eq 9).

Value estimates are related to choices in the same manner used for the RW model (Eq 4).

**Uncertainty model.** To test whether recently experienced uncertainty also influenced choices (in addition to values), we constructed the *Uncertainty* model, updating the recency-weighted uncertainty of each bandit [27] as described in the Results section below (Eq 10).

Note that the PE associated with a particular bandit only gets updated when that bandit was sampled, while the trial counter $t$ refers to all trials. In the *Uncertainty* model, the uncertainties $U$ were then added to the logistic regression:

$$p(k) = \sigma(\beta_0 + \beta_1 V_{l,t} + \beta_2 V_{k,t} + \beta_3 U_{l,t} + \beta_4 U_{k,t}) \tag{5}$$

**Surprise model.** This model asked whether observing surprising outcomes would influence participants' learning rates, compared to observing less surprising outcomes. The core of the model is the variable learning rate $\alpha^*$, which is reported in the Results section (Eq 11). This PE dependent learning rate was used to update values in the standard fashion, i.e.

$$\begin{aligned} V_{k,t+1} &= V_{k,t} + \alpha^* \, \mathrm{PE}_t \\ &= V_{k,t} + \left( l + \frac{2}{1 + \widehat{\mathrm{PE}}^{-s}} \, (u - l) \right) \mathrm{PE}_t \end{aligned} \tag{6}$$

Note that in case $l > u$ the function specifies a decreasing function and v.v. if $u > l$, and the slope parameter allowed to accommodate a wide range of relationships between the learning rate and the predictions error. $\widehat{\mathrm{PE}}$ reflects an absolute PE term defined as

$$\widehat{\mathrm{PE}} = \frac{2}{1 + e^{-0.1|\mathrm{PE}|}} - 1 \tag{7}$$

and scaled by the maximal possible PE value of 60. The updating detailed in Eq 11 altered the estimated values. These values were then used to predict choices as in the baseline model (Eq 4).

**Combined models.** Finally, we tested for the combined influence of uncertainty and prediction-based learning on choices by entering the new values as estimated by either the *Valence* model or the *Surprise* model jointly with the uncertainties into the logistic regression (as in Eq 5) to explain choices. This resulted in two additional models (*Unc+Valence* and *Unc +Surprise*) that jointly considered the influence of uncertainty and the two different models of learning specified in Eqs 11 and 9. We also implemented an additional model that combined the functionality of the Surprise model with that of the Valence model potentially capturing if surprise from positive or negative prediction errors differently affects learning. This model did not provide any superior fits compared to the winning models reported here. Details can be found in the SI.

## Model fitting

Parameter fitting consisted of fitting the $\beta$ coefficients of the logistic choice model, and the parameter(s) of the learning rate function and, if included, the uncertainty function. Fitting minimized the negative log-likelihood of each models' choices using a nested approach akin to a coordinate descent approach, see [48]. Specifically, the parameters of the learning rate function (simply $\alpha$ in the *RW* and *Uncertainty* models, but [$\alpha_{\mathrm{pos}}$, $\alpha_{\mathrm{neg}}$] and [$l$, $u$, $s$] in the *Valence* and *Surprise* models, respectively) were set in an outer loop using a non-linear search method

[49, NLOPT_GN_DIRECT_L] implemented in the nloptr package [50] in R. The $\beta$ coefficients were then set using maximum likelihood estimation in an inner loop, and the resulting choice likelihood was used to inform the non-linear search for the outer parameters.

Importantly, all models were fit solely to participants' free choices in low-mid bandit comparisons. This was done to specifically capture the behavioral effect in response to the one-sided bimodal distribution of the mid bandit. Additionally, since the value estimate of each bandit was initialized with the fixed value of 50, we excluded the first three trials of each bandit from the likelihood calculation to avoid artificially large unsigned prediction errors during initial updating that do not reflect surprise. Guided-choice and estimation trials were not used during the minimization process. Parameters were constrained to lie in the intervals given above. To avoid overfitting, model fits were compared using corrected Akaike Information Criterion (AICc, see [51]) scores, a metric that more strongly considers the amount of trials used.

### Model recovery and posterior predictive checks

Following best practices as specified by Wilson and Collins [52] we quantitatively assessed the model fitting process by conducting full model and parameter recovery (see S1 Text, Figs B and C in S1 Text). Furthermore, a posterior predictive check was performed to validate the winning model (see Fig D in S1 Text). First, we used each model to simulate choice behavior using the reward schedules and fit parameter values of each participant, effectively creating a full data set as if participants behaved perfectly according to each model. To quantify how well the model captures individual decision processes and to assess its plausibility, we repeated the above reported analysis of free choices in low-mid trials before and after the model encountered a surprising outcome of the mid bandit (see Eq 2), but on the artificial data.

### Results

A total of 51 younger (18–30 years, avg.: 24.4) and 51 older (50–73 years, avg.: 57.2) participants performed a value-based choice task online. All details are described in the Methods, but we will briefly repeat the core aspects of our task and models here. The task consisted of two runs á 240 trials in which participants were asked to learn about the value of three bandits, each indicated by a different Hiragana symbol (Fig 1A). Outcomes ranged between 0 and 100 and the averages of the three bandits were set such that one bandit had a low, one a medium and one a high mean, each differing by 16.6 points on average from its neighbor (Fig 1B). Participants learned about the average outcomes through free choice trials in which they could select one out of two offered Hiragana symbols and received an outcome sampled from the corresponding bandit's distribution (192 trials/run, Fig 1A). This produced three trial types, which reflect the pair of bandits participants could choose from: low-mid trials featured bandits with low and medium means; low-high and mid-high trials offered the other respective bandit pairs.

Bandits not only differed in their mean, but also in their distribution. While the low and high bandits had symmetrical Gaussian outcome distributions (SD = 5.55 points), the mid bandit had a bi-modal distribution with a main mode that generated 80% of outcomes and smaller mode that generated 20% of outcomes (Fig 1B). This crucial manipulation allowed us to investigate how sensitive learning was to outcomes that had a large deviation from previously experienced outcomes, but were not the most rare outcomes. To provide enough experience with each bandit's outcome, we asked participants on 20% of trials to select a computer-determined bandit instead of choosing freely (forced choice trials). We also asked participants to directly provide an estimate of each bandit's value using a slider (16 trials / run, Fig 1A, bottom).

## Choices and reaction times in different bandit comparisons

We first asked whether participants learned to make reward-maximizing choices, and whether the proportion of correct responses in free choice trials differed between available options. Task performance of participants is displayed in Fig 2A and was consistently above chance (see Fig A in S1 Text, for additional detail). A corresponding LME model of correct choice probabilities revealed effects of run ($\chi^2(1) = 12.061$, p = .001, post hoc test: t(500) = − 3.473, p < .001) and bandit comparison (low-mid vs. mid-high vs. low-high, $\chi^2(2) = 155.332$, p < .001), but no main effect of age group ($\chi^2(1) = 3.392$, p = .066). This reflected that performance increased across runs (Fig 2B), and that participants performed best on low-high trials (92.6% correct choices), followed by mid-high trials (87.2%) and low-mid trials (79.2%, see Fig 2C). Note that we observed a performance difference between mid-high and mid-low trials (t(500) = 7.420, p < .001) even though the difference in average reward between both bandit pairs was the same, and any value compression for numerically higher outcomes [53, 54] should have the opposite effect (the means in low-mid trials, 30 vs. 50, should be relatively easier to distinguish than the means in mid-high trials, 50 vs. 70). While a standard analysis revealed no group × bandit interaction ($\chi^2(2) = 2.380$, p = .304), this was likely caused by two relatively influential observations in the older age group (see Fig 2D, Cook's distance > 0.1). A direct comparison of the difference between low-mid vs. mid-high trials between older and younger adults using a robust regression approach indicated older participants had a greater performance decrease in low-mid relative to mid-high trials (robust regression with bisquare weights, t(100) = − 2.735, p = .010).

A similar pattern was found in participant's RTs. An LME model of log-transformed RTs also showed a main effect of bandit comparison ($\chi^2(2) = 457.891$, p < .001). Low-mid trials were associated with higher RTs compared to mid-high trials (t(500) = 16.121, p < .001), while the fastest RTs were observed in low-high trials (mid-high vs. low-high: t(500) = 4.126, p < .001). The same LME model also showed a main effect of age group ($\chi^2(1) = 31.356$, p < .001) as well as an age group × bandit comparison interaction ($\chi^2(2) = 16.360$, p < .001). This reflected that older adults in general reacted slower than younger adults (t(100) = 5.600, p < .001) and that age differences in log-transformed RTs were more pronounced in low-mid trials (age difference = .236, t(123) = 6.589, p < .001) compared to mid-high trials (age difference = .164, comparison of age differences between low-mid and mid-high: t(500) = − 3.639, p < .001; see Fig 2F) and low-high trials (t(500) = − 3.349, p < .001).

## Effects of highly surprising events on subsequent choices

To better understand how participants were influenced by surprising outcomes, we investigated the immediate effect of outcomes that elicited large absolute PEs in the low-mid bandit on subsequent choices. Comparing the proportion of mid bandit choices in low-mid trials immediately before vs. after a surprising outcome (below 20th percentile of mid bandit, see Methods) revealed that older participants chose the mid bandit less often following large absolute PEs, which occurred when outcomes where drawn from the second mode of the mid bandit's bi-modal distribution (pre-post difference in older adults: 17.6%; t(299) = 5.458, p < .001, see Fig 3A). This was less the case for younger adults (6.6%; t(299) = 2.052, p = .080, Šidák corrected; position × age group interaction: $\chi^2(1) = 5.844$, p = .016; main effect pre vs. post across age groups: $\chi^2(1) = 28.160$, p < .001; 82.2% vs. 70.1%). The model of the immediate effect of surprising outcomes also indicated a significant main effect of run ($\chi^2(1) = 12.438$, p < .001) that reflected a general increase in mid bandit choices in low-mid trials in the second run (t(299) = − 3.530, p < .001). No evidence of a main difference between age groups in mid-bandit choices on average was found ($\chi^2(1) = 1.363$, p = .243). Given that risk aversion would lead

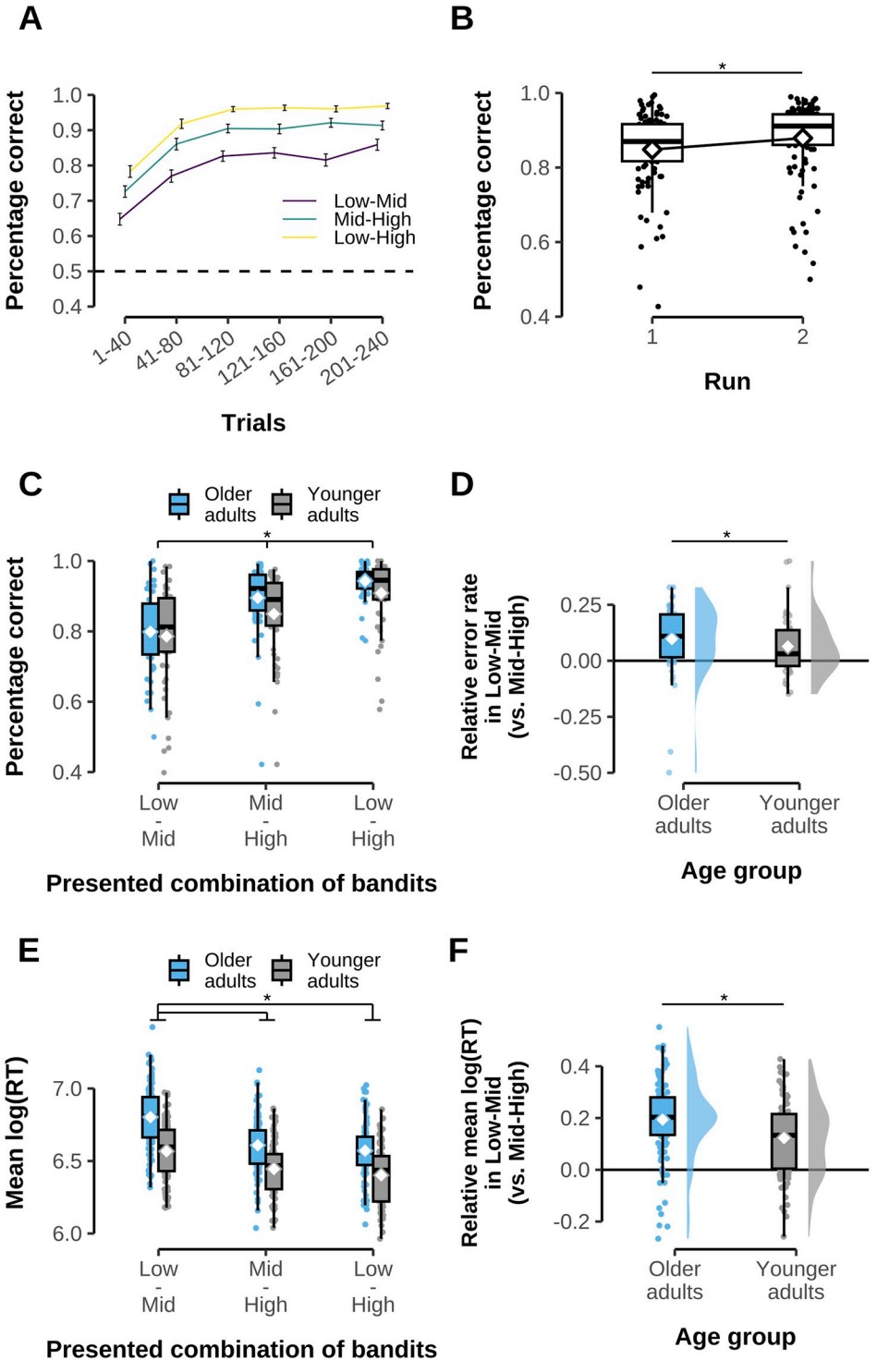

**Fig 2. Task performance. A**: Percentage of correct free choice trials (i.e. choosing bandit with higher average outcome, y-axis) over trials within run (bins of 40, x-axis). Data shown separately for each condition/bandit combination (low-mid, mid-high, low-high, see colors/legend). Dashed line indicates chance-level performance and error bars show standard error of the mean. **B**: Average free choice accuracy for each run, irrespective of presented bandit combination. Each dot corresponds to one participant. **C**: Proportion of correct responses on free choice trials across runs, separately for condition/bandit combination and for older (blue) vs. younger adults (grey, see legend). White diamonds represent group averages, each dot one participant. **D**: Difference between error rates in the low-mid vs. mid-high trials, separately for each age group. Values larger than zero indicate more errors in low-mid trials compared

to mid-high trials. Colors as in panel C. White diamonds represent group averages, each dot one participant. Note that average reward of the mid bandit was equidistant to the low and high bandits. **E**: Average log reaction times on free choice trials across runs, separately for condition/combination of bandits (low-mid, mid-high, low-high) and for older (blue) vs younger adults (grey, see legend). White diamonds represent group averages, each dot one participant. **F**: Difference between log RTs in the low-mid vs. mid-high trials, separately for each age group. Values larger than zero indicate slower responses on low-mid trials compared to mid-high trials. Colors as in panel C. White diamonds represent group averages, each dot one participant. Stars and lines indicate significant statistical tests as described in the text. Note, this does not include interaction effects to avoid clutter.

participants to avoid the more variable mid bandit in general, it appears that risk aversion (or age differences therein) cannot explain the result reported above. We also checked whether the age difference could reflect a general reaction towards below-average outcomes of the mid bandit by repeating the same analysis for low-mid trials immediately before and after less surprising outcomes (mid-bandit outcomes between the 20th and 40th percentile). No effects of either pre vs. post or age group were found ($\chi^2(1) = .541$, p = .462 and $\chi^2(1) = .968$, p = .975, respectively, see Fig 3B).

## Value estimates

We next checked for systematic biases in estimation trials by asking how accurate participants' value estimates were relative to the ground truth running average of each bandit. To avoid effects of non-stationarity during early learning, analyses were restricted to the second half of each run. Comparing the across-run average estimates vs. ground truth separately for each

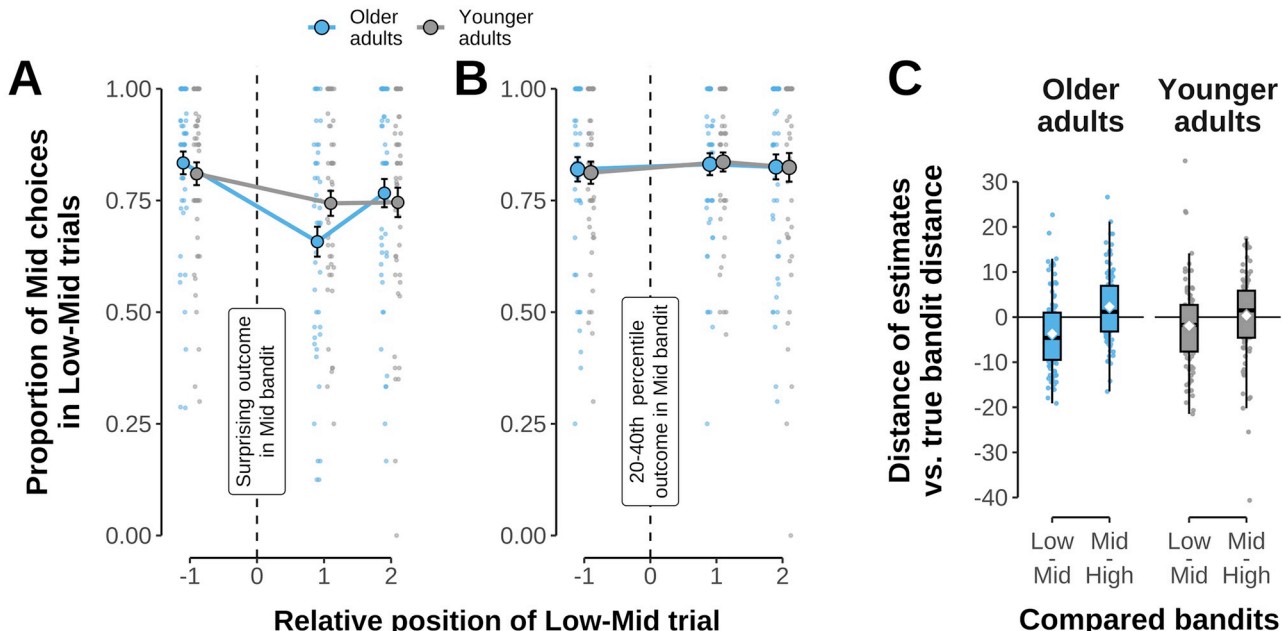

**Fig 3. Influence of surprising outcomes on choice and value estimation. A**: Proportion of mid bandit choices in low-mid trials one trial before and two trials after participants experienced a surprising outcome from the mid bandit (vertical dashed line). Data shown separately for older (blue) and younger adults (grey). Each small dot is one participant, large dots depict group means with standard error of the mean shown by error bars. **B**: Same analysis, but now time-locked to trials in which moderately low outcomes of the mid bandit (20th to 40th percentile) were experienced. Colors and other details as on left. **C**: Difference between reported and true average outcome difference between bandit pairs, separately for both age groups. Reported values reflect participant responses on estimation trials, while true bandit distances were calculated as the difference between the running means of experienced outcomes until the time of a given estimation trial. Values lower than zero indicate an underestimation of bandit distance, perceiving them closer than their true numeric distance. Colors and dots as in panel A and Fig 2.

bandit and age group did not indicate any significant difference (younger: ps ≥ .111, older: ps ≥ .625; the largest difference found indicated a non-significant underestimation of the high bandit in younger adults t(48) = − 2.437, p = .111, Bonferroni-Holm corrected). This indicates that value estimates of older and younger participants were accurate and unbiased on average.

Notably, we also found that older and younger adults showed selective biases in how well they estimated the value of the mid bandit relative to either the high or low bandit. Specifically, comparing the difference in ratings between each bandit pair to the true experienced difference (difference of running means) revealed older adults perceived the low and mid bandit to be closer together compared to their true distance (they underestimated the difference by − 3.68 points), more than younger adults did (–1.93 points; comparison of underestimation of low vs. mid bandit between age groups: t(277) = 2.051, p = .041; interaction bandit pair × age-group: $\chi^2(1) = 8.280$, p = .004; main effect bandit pair: $\chi^2(1) = 34.536$, p < .001, see Fig 3C).

## Computational modeling

Above we reported that following mid-bandit outcomes with large prediction errors, participants shifted choice preferences away from the mid bandit in a manner that suggests heightened learning from such singular events. Correspondingly, participants performed worse in low-mid trials compared to mid-high trials, and underestimated the same bandit differences. This effect was particularly pronounced for older adults.

Building on these behavioral results, we used computational modeling to specifically contrast the contributions of surprise, uncertainty and differential learning from positive and negative prediction errors (as well as combination of these) to behavior. While the behavioral effect following surprise trials reported above is qualitatively consistent with our hypothesized mechanism, computational models allow us to test a more precise version of our hypothesis across the entire sequence of choices. We therefore modeled participants choices specifically in low-mid trials (see Methods) using the following four main and two combination models:

A *Rescorla-Wagner* model that assumes a constant learning rate and no effects of uncertainty or surprise (Fig 4A), served as a baseline. Learning in this model followed a standard delta rule:

$$
\begin{aligned}
V_{k,t+1} &= V_{k,t} + \alpha \left( R_{k,t} - V_{k,t} \right) \\
&= V_{k,t} + \alpha \, \mathrm{PE}_t,
\end{aligned}
\tag{8}
$$

where $V_{k,t}$ denotes the value estimate of bandit $k$ on trial $t$, and $R_{k,t}$ is the corresponding reward obtained at time $t$ after choosing $k$. The difference between the expected and obtained value is referred to as the prediction error (PE, here $R_{k,t} - V_{k,t}$), and $\alpha \in [0, 1]$ is a learning rate fitted as a free parameter.

The *Valence* model (Fig 4C) assessed if participants choices were best explained by differential learning from positive versus negative PEs, c.f. [44], which previously has been observed in the context of aging [35, 47]. This model was also necessary given that the bimodal distribution of the mid bandit led to predominantly large negative PEs. Hence, we modified Eq 8 to include two separate learning rates $\alpha_{\mathrm{pos}}$ and $\alpha_{\mathrm{neg}}$ for positive and negative prediction errors, respectively:

$$
V_{k,t+1} = \begin{cases} V_{k,t} + \alpha_{\mathrm{pos}}\mathrm{PE}_t, & \text{if } \mathrm{PE}_t \geq 0 \\ V_{k,t} + \alpha_{\mathrm{neg}}\mathrm{PE}_t, & \text{if } \mathrm{PE}_t < 0 \end{cases}
\tag{9}
$$

The *Uncertainty* model (Fig 4B) tested whether participants' choices were influenced by the unsigned magnitude of past prediction errors, i.e. whether they showed less or more

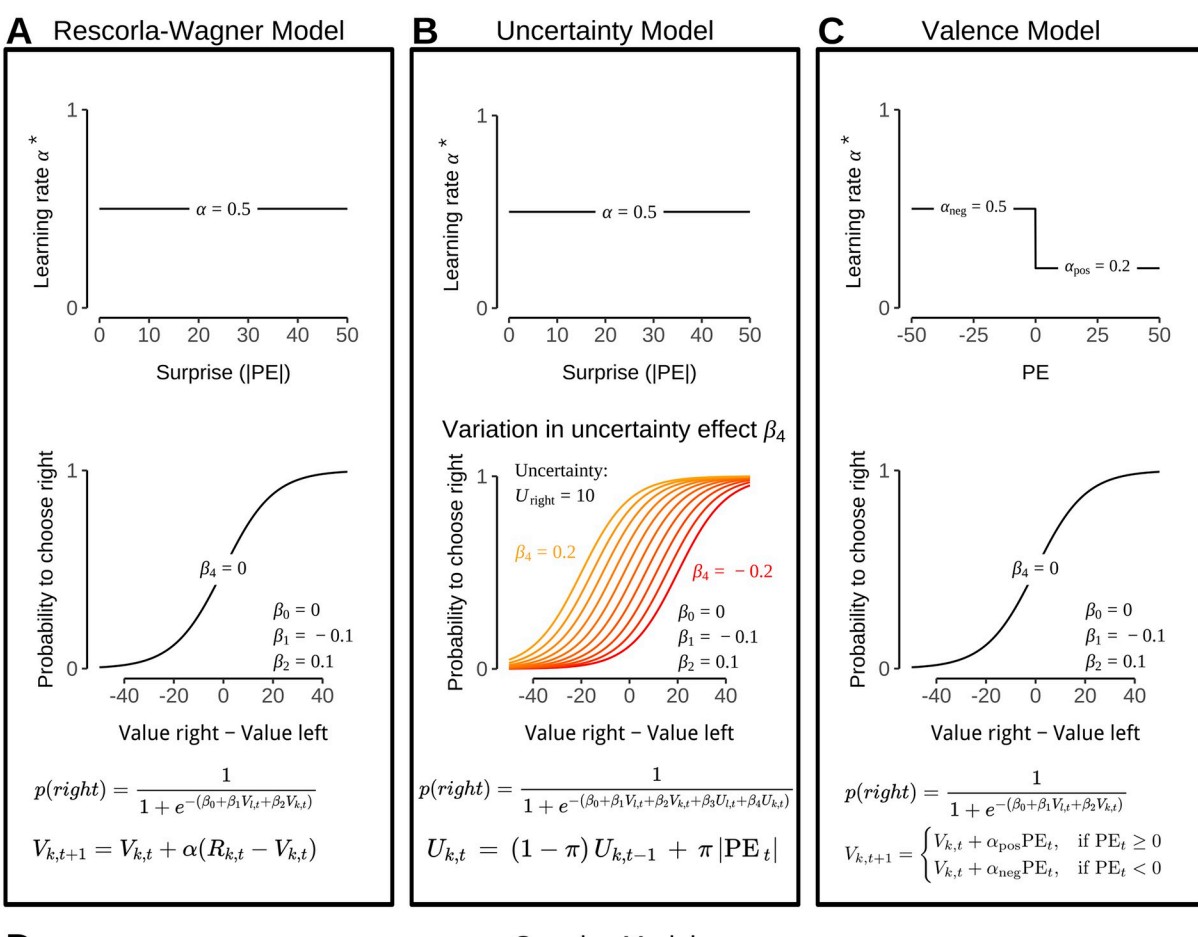

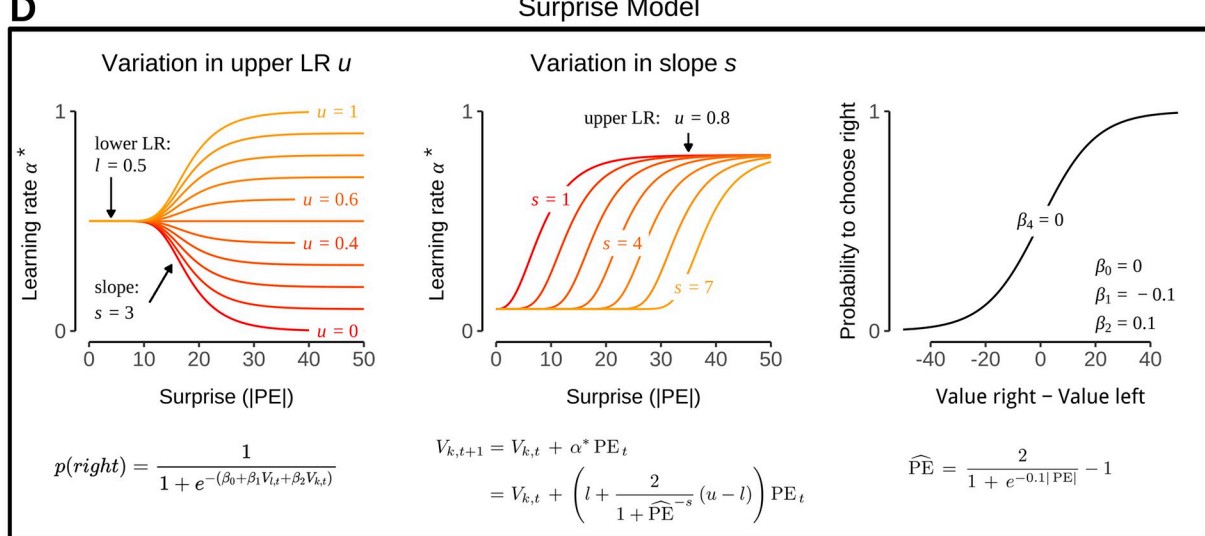

**Fig 4. Illustration of computational models.** For each model depicted is the sensitivity of trial-wise instantaneous updates (learning rate) to the surprise (i.e., unsigned prediction error) associated with an outcome of a bandit choice. For the *Valence* model (panel B) this is shown relative to the signed prediction error to display different learning from outcomes lower or higher than expected. Further, via $\beta_4$ is shown the influence of the right bandit's uncertainty ($U_k$, estimated by the agent) on choice probability of the right bandit (see Eq 4). **A**: *Rescorla-Wagner* model in which updates and choices are insensitive to both, surprise and uncertainty. **B**: *Uncertainty* model in which updates are insensitive to surprise but bandit choices are influenced by uncertainty. Note, how uncertainty in the right bandit can heighten ($\beta_4 > 0$) or lower the probability of choosing the right bandit ($\beta_4 < 0$). Uncertainty estimate of the right bandit is fixed to $U_k = 10$ for the illustration but in the model depend on a free parameter $\pi$ (see Eq 10). The influence of the left bandit's uncertainty ($\beta_3$) is left out for simplicity. **C**: *Valence* model which is

insensitive to surprise and uncertainty but allows for different strength of instantaneous updates depending on the sign of the prediction error. **D**: *Surprise* model which is insensitive to bandit uncertainty, but in which trial-wise updates are influenced by surprise in dependence of the parameters *l*, *s*, and *u* (see Eq 11). High levels of surprise can either increase ($u > l$) or decrease the learning rate on a given trial ($u < l$), here shown by a variation in *u* (first graph). A similar variation of the *l* parameter is left out for simplicity. Lower values of the slope parameter *s* indicate that updating is adjusted already for lower levels of surprise (second graph). Not depicted are the *Uncertainty+Surprise* and *Uncertainty+Valence* models which combine the principles of **B** with those of the **C** and **D**, respectively.

preference for bandits that were associated with high/low uncertainty in the past. We adopted the associability implementation used by Li et al. [27] to keep track of the recency-weighted uncertainty of each bandit:

$$U_{k,t} = (1 - \pi) U_{k,t-1} + \pi |\mathrm{PE}_t| \tag{10}$$

The free parameter $\pi \in [0, 1]$ determines the degree of recency-weighting of prediction errors that form the agent's current uncertainty estimate. Values close to 1 mean that uncertainty estimates are driven by recent outcomes while vales closer to 0 mean that the agent considers a long history of errors.

The fourth main model was the *Surprise* model (Fig 4D), which assumed that values are learned with a learning rate that depends on the amount of surprise in a given trial. The core idea of this model, compared to the Rescorla-Wagner and Uncertainty models, was that how much change in value results from a particular outcome depends on the *absolute* prediction error. The model was designed to incorporate various relationships between absolute prediction error and learning rate, including both higher learning rates for low prediction errors and the opposite. Moreover, we assumed that the effect of prediction error on learning rate is instantaneous, i.e. affects updating on the trial immediately, in contrast to the Uncertainty model, where prediction errors on trial *t* only come to influence the learning rate on trial *t* + 1. To this end, we modified the learning rule given in Eq 8 to include a variable learning rate $\alpha^*$, which itself was a logistic function of the scaled absolute prediction error $\widehat{\mathrm{PE}}$ (see Methods),

$$\alpha^* = l + \frac{2}{1 + \widehat{\mathrm{PE}}^{-s}} (u - l) \tag{11}$$

and updating followed the same delta rule given in Eq 8, using $\alpha^*$ instead of $\alpha$ (see Methods, Eq 6). The introduction of $\widehat{\mathrm{PE}}$ in the equation of $\alpha^*$ was necessary to achieve rescaling into the range of [0, 1], which is needed for learning rates. The Surprise model included three free parameters that regulated the influence of prediction error dependent surprise: a lower bound $l \in [0, 1]$ that specified the alpha level when the PE was 0, a upper bound $u \in [0, 1]$ that determined the learning rate when the PE was maximal, and a slope, $s \in [1, 7]$ between these two extremes. Fig 4D illustrates the behavior of this flexible learning rate function under some parameter variations.

Finally, two additional models, *Uncertainty+Valence* and *Uncertainty+Surprise*, combined the uncertainty mechanism with the updating mechanisms of the *Valence* and *Surprise* models, respectively. The four main models are illustrated in Fig 4. The model predictions were related to choices using a logistic regression approach and parameters were fit using Maximum Likelihood estimation (Details see Methods). Results of an additional model that combined the functionality of the Surprise and Valence models can be found in the SI. To fit participants' choices, value and uncertainty estimates were included in a logistic choice model as predictors. The corresponding $\beta_{1-4}$ parameters represent the influence of value/uncertainty of left and right bandits on choice (See Methods).

Basic analyses indicated that the RW account of value learning captured core aspects of participant behavior (see Methods). Yet, the *RW* model did not offer the best fit to participants' choices. Instead, the model comparison showed that the corrected AIC score (AICc, [51]) was lowest for the *Surprise* model (AICc: 115.40 vs. 119.56 of the RW model). The next best models were the *Valence* (115.76), *Uncertainty* (116.08), *Unc+Valence* (116.11), and *Unc+Surprise* (116.86) models (see Fig 5A, AICc of *RW* model: 119.56). A protected exceedance probability analysis showed the *Surprise* model as the most likely data-generating process across all participants (77.0%; via R package `bmsR` [55]; $10^5$ samples; see [56]), followed by the *Valence* model (22.7%, see Fig 5B). Further analyzing the participant-wise differences in AICc scores separately for each age group showed that, while the *Surprise* and *Valence* models had significantly lower AICc scores compared to the *RW* model in both age groups (Surprise Model—older: t (50) = –3.39, p = .011, younger: t(50) = –3.44, p = .011; *Valence* model—older: t(50) = –3.10, p = .022, younger: t(50) = –3.53, p = .009), this was not the case for the *Uncertainty* or combined models (all t(50) > –2.08, ps ≥ .254, all ps Bonferroni-Holm corrected). Finally, using participant-wise AICc comparisons to identify the winning model within each participant (see Fig 5C) also identified the *Surprise* model as the most frequently winning model across all participants (30 participants followed by the *RW*, *Valence* models, see Fig 5C).

We next performed posterior predictive checks to ask whether the two models with the highest protected exceedance probability (*Surprise* and *Valence* model) showed the main behavioral observation of interest, i.e. the outsized effect of large absolute PE events on choices (see Methods). We used the estimated models to generate synthetic data for each model. We then analyzed the generated data sets in an identical manner to the participant's data. A graphical comparison can be found in Fig D in S1 Text. Both models showed a significant main effect of pre vs. post large absolute PEs ($\chi^2(1)$ = 9.01, p = .003 and $\chi^2(1)$ = 7.30, p = .007 for *Surprise* and *Valence* model, respectively), as found in participant behavior. An LMM of the pre- vs. post change across age groups indicated a marginally significant age group × model interaction ($\chi^2(1)$ = 3.53, p = .06). Post-hoc tests of this interaction showed that the *Surprise* model qualitatively captured the pattern of results, i.e. older adults showed a larger adaptation than younger adults (estimate of older vs. younger contrast in *Surprise* model: .053). This was not the case for the *Valence* model, where the effect was reversed (i.e. younger adults adapted more to large PE events than older adults, –.046). Despite this qualitative match of the age differences between the real data and the *Surprise* model, the corresponding post-hoc comparison did not reach significance (t(189) = 1.23, p = .219; the same was true for the *Valence* model, t (189) = –1.08, p = .281). Correspondingly, neither model captured the significant time-point × age group interaction evident in the behavioral analysis ($\chi^2(1)$ < 1.55, p > .213).

Given that the evidence overall favored the *Surprise* model, we lastly analysed its parameters in more detail. In a first sanity check, we confirmed that the $\beta_1$ and $\beta_2$ parameters, reflecting the influence of estimated left and right bandit value on choices correlated with performance as expected ($\beta_1$: r = –.36, t(100) = –3.88, p < .001; $\beta_2$: r = .35, t(100) = 3.74, p < .001, Bonferroni-Holm corrected). We then investigated the parameters related more specifically to the surprise effect on learning rate (*l*, *u*, and *s*; see Eq 11). Results are shown in Fig 6. A core interest was on the difference between the upper and lower bound parameters, where positive values ($u - l > 0$) reflect relatively faster learning from large prediction errors while negative values ($u - l \leq 0$) reflect relatively faster learning from small PEs. Surprisingly, however, we did not find any relationship between the main behavior proxy of surprise sensitivity discussed above (Fig 3A) and the parameter difference $u - l$ (r = –.093). Moreover, the parameter difference did not indicate any average bias to learn more or less from surprising outcomes, i.e. $u$—$l$ was not significantly different from 0 in either age group (younger: $u - l$ = –0.10, t(50) = –1.33, p = .380, older: $u - l$ = 0.00, t(50) = .020, p = .984, one-sided t-test against 0, Bonferroni-Holm

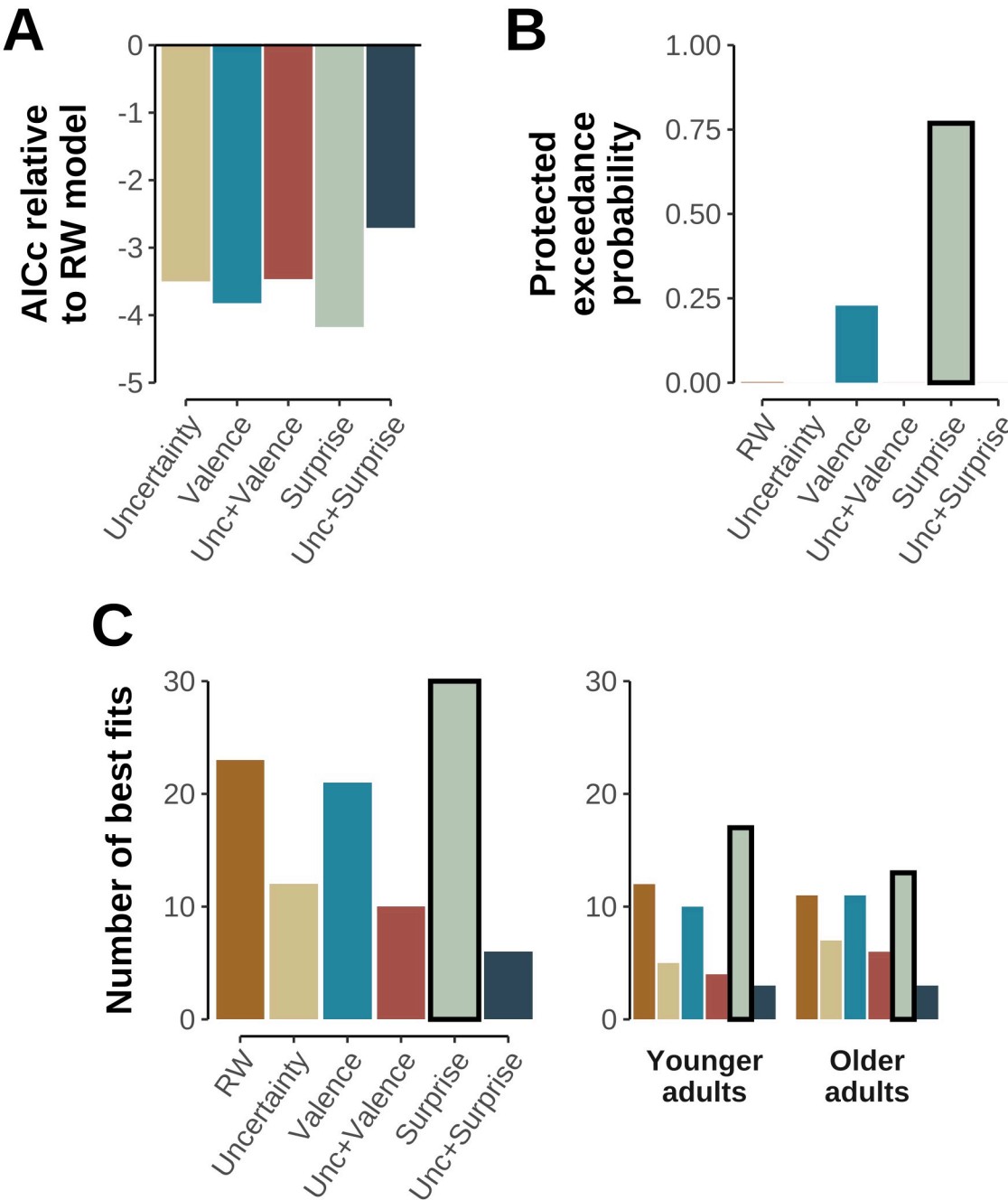

**Fig 5. Model comparison. A**: Average difference of AICc scores between candidate models and *RW* model (AICc$_i$—AICc$_{RW}$). Lower values indicate a better fit compared to the *RW* model. Note, that we found substantial interindividual variability in AICc scores, see panel C and text. **B**: Protected exceedance probability across all six candidate models and across both age groups. See [56] for details. **C**: Number of participants for which each model had the lowest AICc shown across (left) and within age groups (right).

corrected, see Fig 6A). Splitting participants into groups based on whether the learning rate was decreased ($u − l < 0$) or increased ($u − l > 0$) for large absolute PEs showed that numerically more older than younger adults had increased learning rates (25 vs. 18), but a formal analysis did not indicate any significant difference in age groups ($\chi^2(1) = 1.45$, p = .229, see Fig

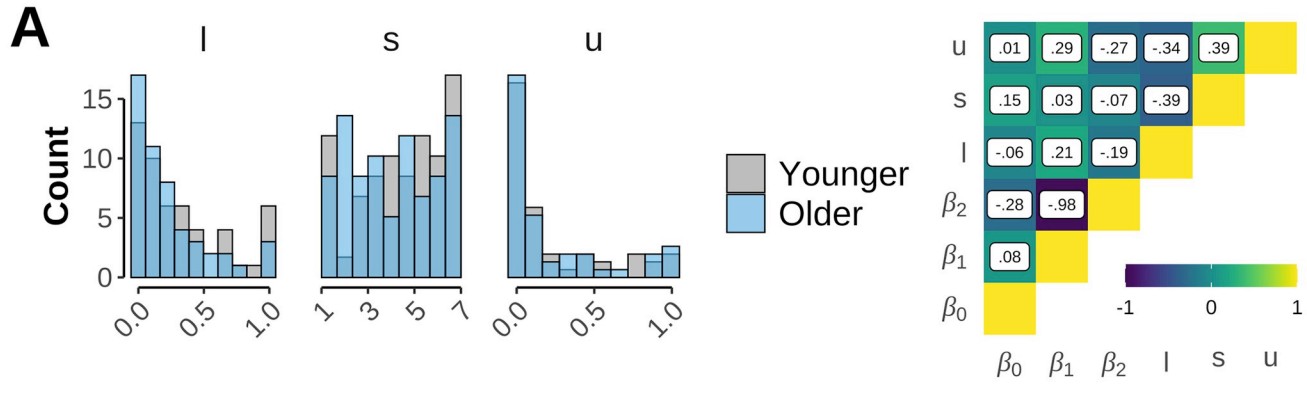

**Fig 6. Analysis of *Surprise* model. A**: Distribution and correlation between *Surprise* model parameters. On left: Histogram of model parameters *l*, *s*, and *u* involved in instantaneously adjusting learning rate as a function of surprise (i.e. unsigned prediction error; see Eq 11) for younger and older adults. On right: Correlation matrix between all model parameters. Brighter colors show stronger positive correlation, darker colors stronger negative correlation. In each cell is shown the Pearson correlation between the respective parameters. **B**: Individual relationships between surprise about outcome (i.e., unsigned prediction error) and trial-specific learning rate $\alpha^*$ as specified by the model parameters *l*, *s*, and *u* (see Eq 11). Depicted separately within each age group are participants whose updating for high levels of surprise is *decreased* ($u - l < 0$, left) or *increased* ($u - l > 0$, right). **C**: Number of participants within each age group showing decreased ($u - l < 0$) or increased ($u - l > 0$) updating from higher levels of surprise. **D**: Age group comparison for parameters specifying differential updating from surprising outcomes. On left: Difference between *u* and *l* parameter. Values above zero indicate increased updating

from surprising outcomes. Dots show individual values, diamonds show group-specific mean. Depicted in the middle are density plots of the respective age group's parameter distribution. On right: Slope parameter $s$. Depiction identical to left plot.

6C). A direct comparison of $u - l$ values between both age groups did not reveal any age differences (t(99.65) = –0.98, p = .330, see Fig 6D). Likewise, the slope parameter $s$ did not differ significantly between age groups (t(99.98) = 1.07, p = .285, see Fig 6D).

Taken together, the computational modeling results suggest that participants' choices in low-mid bandit comparisons were mainly driven by surprise, as opposed to other candidate mechanisms such as uncertainty, differential learning from positive or negative prediction errors, or their combination. This was shown by the best fit of the *Surprise* model. Posterior predictive checks indicated that the *Surprise* model also was best at qualitatively reproducing the behavioral proxy of surprise-triggered learning, and age differences therein. However, we also found substantial between-participant variation in model fits, and analyses of the *Surprise* model's parameters did not show any age differences related to the influence of surprising outcomes in low-mid trials evident in the behavioral analyses. This suggests that on top of a main age difference in surprise dependent learning, behavior in our task reflects a complex mixture of different computational strategies engaged by both older as well as younger adults.

## Discussion

In this study we investigated over- and underweighting of surprising outcomes during reinforcement learning, and asked whether age differences exists in this process. Our main hypothesis was that older adults show greater sensitivity to outcomes that elicit large absolute prediction errors compared to younger adults. To this end, we analyzed behavior of 51 younger and 51 older participants in a multi-armed bandit task featuring two bandits with a Gaussian reward distribution of low and high mean, and one bandit with an asymmetric, bi-modal reward distribution of intermediate value. The asymmetric nature of the mid bandit's reward distribution was designed as such that overweighting of surprising outcomes during learning should result in non-optimal choices when comparing the mid-value (i.e., bimodal) bandit to the low-value bandit. We found that behavioral accuracy in low-mid bandit choices was significantly lower compared to mid-high trials despite the fact that both bandit pairs exhibited the same difference in their mean outcome. This was particularly the case for older compared to younger adults. This suggests that surprising outcomes are overweighted, relative to ordinary outcomes, and that this effect becomes more prominent with age. This effect was also present in explicit value ratings, in which both age groups underestimated the difference in average rewards of the low and mid bandit, and older adults showed a stronger tendency to do so. An analysis of detailed choice time courses also found that surprising outcomes had a stronger influence on consecutive choices in older adults compared to younger adults, suggesting a greater sensitivity to surprising outcomes in older adults. To explain these findings more formally, we compared six RL models that allowed us to address if participants' choices in low-mid bandit comparisons were driven by either uncertainty, differential updating from positive and negative prediction errors, scaled learning from more surprising outcomes, or a combination of both. Model comparison indicated that the *Surprise* model offered the best explanation of participants' decisions overall. Yet, a closer inspection also revealed that our data was characterized by large interindividual variability in the model that best explained different participants' data, and that fitted model's parameters did not reflect the age differences evident in the behavioral analyses. We suspect that these findings are partly caused by issues of model identifiability, as evidenced by the reduced model recovery (see Fig C in S1 Text).

Our results might appear inconsistent with findings that older adults show stronger over-weighting of low probability events (i.e., more risk-taking) when confronted with gambles in the gain and mixed domain [20]. Inspired by cumulative prospect theory [13] and using a risky decision making paradigm [5], Pachur and colleagues [20] asked participants to choose between two binary monetary lotteries. Their core finding was that in decisions from description, older adults overestimated the probability of rare events more than younger adults. However, there are at least three crucial differences between Pachur et al. [20] and our study that might explain these seemingly contradictory results. First, we examined the domain of decisions from experience, in contrast to description-based choices. We found that older participants were sensitive to high prediction error events, which in our case resulted in higher risk aversion. Second, our results suggest that during learning, older adults do not exhibit heightened sensitivity to rare events per se, but rather to events that elicited particularly large prediction errors (i.e. surprise, which did not play a major role in Pachur et al. [20]). Finally, in our case outcomes can less clearly be categorized as gains or losses. That is, although we did not include punishment per se, participants' choices were often governed by negative prediction errors arising from less positive outcomes than expected.

Recently, another body of work modeled the influence of extreme/surprising events on decisions using sequential sampling tasks [57, 58] which are more closely connected to the kind of instantaneous trial-by-trial updating we investigated in this study. These tasks present participants with repeated samples in quick succession from one or more distributions, including extreme samples from the distributions edges, and ask for a judgment of the mean over samples. The results support the idea of selective weighting of extreme outcomes also in the context of a task that is based on a sequential learning process. We believe that also this work is extended by our findings. In particular, we show that similar behavioral patterns emerge also in much slower, single-trial sampling rates and trial-wise choices.

Previous work has also shown that the probability of events that come to mind easily tends to be overestimated [59, *availability bias*], and that memory for values at the edges of distributions is better [15–17]. This might explain the distorted probability weighting functions described above. Note, however, that although in our task the low, mid and high outcome distributions differed in their standard deviation, bandits in our task exhibited similar amounts of low-probability outcomes (see Fig 1B). Hence the choice between bandits was not conflated by choices between a safe and risky gamble as characterized by different probability profiles. Our work does speak directly to the above mentioned memory biases, and suggest the here reported age-differences could be mediated by memory for extreme outcomes.

Our study is also related to work that has focused on how learning rates are adapted in non-stationary environments, in which the true value of bandits changes over time [31, 60, 61]. Unlike our own experiment in which participants learn from bandits with a stable outcome distribution, most of these studies investigated how participants infer the environment's uncertainty and volatility (rate of change), and adapt their learning rates in response to these variables. Our *Surprise* model differs substantially from these accounts in that it assumes no computation of volatility or uncertainty for future use. Rather, the model captures the possibility of instantaneous increase in learning rate when outcomes elicit large prediction errors, with no effects on subsequent learning rates, as would be predicted by uncertainty or volatility-based accounts. Thus, our models are most informative for understanding if surprising events get treated differently in reward-based learning in stationary environments. Most relevant for our study is research that investigated the effects of age on the role of uncertainty and surprise in learning [3]. In this present work, uncertainty was operationalized as a recency-weighted average of absolute prediction errors. According to previous normative accounts, uncertainty should be the dominant driver of learning rates in stationary environments.

Surprise, in contrast, captures the immediate effect of an unexpected outcome, i.e. the unsigned prediction error. Although the previous work discussed above has largely pointed out that older adults tend to underestimate uncertainty [3], it also found some evidence to suggest that in response to surprise older adults adjusted their learning rate more than younger adults. More recent results extend these insights [21], suggesting that older adults often even completely ignore prediction errors attributable to outcome uncertainty, while showing similar surprise sensitivity as their younger counterparts. In contrast to these studies, our work examines surprise-driven learning in stationary environments, where surprise should have more subtle effects on learning, suggesting that in these circumstances, older adults show heightened sensitivity.

Computational modeling of participants' behavior was in line with the idea that surprising events are treated differently during learning. A model that allowed for altered (i.e. increased or decreased) updating from surprising events offered the best prediction of participants' choices in low-mid bandit comparisons. One limitation of this work was the fact that model parameters did not reflect age differences evident in behavioral analyses. Specifically, the model did not suggest a heightened sensitivity to surprising outcomes that is more pronounced in older adults. A potential reason for this finding might be that the effects of surprising outcomes on participants choices can only be reflected in a limited number of trials, reflecting a problem inherent in the study of surprising events. This holds the potential danger of model fits that are largely dominated by behavior in which the differential effects of surprise cannot be reflected in participants' choices. To counteract this, we made the likelihood of each model only dependent on the key comparison regarding surprising outcomes. One additional way to address this issue could be to increase the number of bandits in the task that allow for large prediction errors. This might, however, lead to increased task difficulty. Due to the online setting of the task we decided for a more simple paradigm but results have shown that older as well as younger adults perform adequately on the task. Increasing task difficulty in favor of a more fine-grained characterization of the effect of surprising events on choices therefore seems feasible. This could also help the model identifiability. The model recovery test (see S1 Text; Fig C in S1 Text) showed that in the current task choices simulated from more complex models (e.g. the *Surprise* model) are sometimes attributed to the *RW* model. This is likely explained by the fact that the *RW* model is a special case of all other, more complex models (for instance when $u = l$, the *Surprise* model and the *RW* model are identical), but requires a smaller number of free parameters. By increasing the complexity of the task a larger set of critical trials could be used to make computational models of participants' choices more distinguishable. This would further help to improve the understanding of choice mechanisms in the context of surprising events and how they change with age.

There are also additional considerations that concern the fact that our data was collected online via Prolific. The Prolific platform was specifically build to conduct research and requires comprehensive profiles of the participants [62], and thus represents an adequate choice to collect data also for older adults. Nonetheless, concerns may be raised regarding how representative older age group on Prolific is. Skilled internet use of older adults is more common in populations with higher income and education [63] as well as better levels of health and activity [64]. It is therefore likely that the online data collection sampled a slightly different, high-performance population of older adults when compared to the population sampled in an offline setting. In line with this, our data did not show evidence for age differences in general performance, although older adults tend to perform worse on reward-based learning tasks [7, 65]. Furthermore, less control of the experimental environment can lead to increased noise, reflected for instance in lower learning performance [66]. Since it is possible that an

offline setting might lead to more pronounced age differences in our analyses, it would be beneficial to repeat the same experiment in an in-lab setting.

Taken together, we found behavioral patterns suggesting that overweighting of surprising events was stronger in the group of older adults. A model that instantaneously adjusted learning rates based on the surprise of the experienced outcome explained key choices (low-mid bandit trials) better than other candidate models, including an uncertainty model, and helped to establish an understanding of the learning from surprising events in the context of stationary outcome-based learning. However, since the model parameters fell short of explaining the behavioral age differences, future research should aim to more clearly identify if surprise-related alterations of learning present a general mechanism in the context of stationary environments, or a principle that only gets applied locally to outstanding outcomes and see if the model at hand can be improved to accurately mimic the found behavioral choice patterns. This study provides insight into the differential weighting of surprising events during a reinforcement learning task and, more generally, the role of aging in human decision making.

## Supporting information

**S1 Text. Supporting text.** Supplementary information file including additional analyses of parameter recovery, model recovery and the combined Valence and Surprise model. This file includes Fig A (Performance early in the task for each age group), Fig B (Parameter recovery), Fig C (Model recovery.) and Fig D (Posterior predictive check). Figure legends see inside S1 Text.
(PDF)

## Acknowledgments

We thank all members of the Schuck Lab for their valuable input on our work, and Faculty of the IMPRS LIFE, especially Ulman Lindenberger, for input on our project.

## Author Contributions

**Conceptualization:** Christoph Koch, Ondrej Zika, Nicolas W. Schuck.

**Data curation:** Christoph Koch.

**Formal analysis:** Christoph Koch, Nicolas W. Schuck.

**Funding acquisition:** Nicolas W. Schuck.

**Investigation:** Christoph Koch.

**Methodology:** Christoph Koch, Ondrej Zika, Rasmus Bruckner, Nicolas W. Schuck.

**Project administration:** Nicolas W. Schuck.

**Resources:** Nicolas W. Schuck.

**Supervision:** Nicolas W. Schuck.

**Visualization:** Christoph Koch.

**Writing – original draft:** Christoph Koch, Ondrej Zika, Nicolas W. Schuck.

**Writing – review & editing:** Christoph Koch, Ondrej Zika, Rasmus Bruckner, Nicolas W. Schuck.

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
