## [Decision Letter · Decision Letter 0]

23 Jan 2024

Dear Dr. Schuck,

Thank you very much for submitting your manuscript "Influence of surprise on reinforcement learning in younger and older adults" for consideration at PLOS Computational Biology.

As with all papers reviewed by the journal, your manuscript was reviewed by members of the editorial board and by several independent reviewers. In light of the reviews (below this email), we would like to invite the resubmission of a significantly-revised version that takes into account the reviewers' comments.

As you can see from the below reviews, both reviewers appreciated the study design and the results, but have a few concerns regarding the task and the analysis. We would like to see all concerns addressed in a revised manuscript.

We cannot make any decision about publication until we have seen the revised manuscript and your response to the reviewers' comments. Your revised manuscript is also likely to be sent to reviewers for further evaluation.

Sincerely,

Tobias U Hauser, PhD

Academic Editor

PLOS Computational Biology

Thomas Serre

Section Editor

PLOS Computational Biology

Reviewer's Responses to Questions

**Comments to the Authors:**

Reviewer #1: In this paper by Koch et al., the authors employ a two armed bandit task with Gaussian rewards in combination with computational modeling to test the hypothesis that older adults are more sensitive to surprising outcomes when making decisions under uncertainty. The behavioral task is designed to decouple surprise, the main variable of interest, from reward magnitude, probability and uncertainty. The main claim of the paper is that older adults are more sensitive to surprising events than younger adults. The behavioral analysis reveals interesting age-related behavioral effects that support the claim, but are not recapitulated by the computational modeling analysis. I found the task design and analysis to be careful, and the paper balanced in its claims. The main weakness is in the discrepancy between behavioral and computational modeling results. Below are some comments and suggestions that may address this issue.

Major comments:

- The bimodal distribution of the mid-bandit results in predominately negative large prediction errors – so taking the absolute value and saying the surprise is “valence free” is not truly accurate as a majority of these are actually negative surprise. Why wasn’t an additional condition included that induced large predominately positive PEs?

- Relatedly, have the authors considered a way to combine the "Valence" and "Surprise" model? It is possible that learning from negative outcomes might be impacted differently by surprise than learning from positive outcomes.

- The "Valence" model has some additional machinery to handle the large negative surprises that the RW model does not, which explains why it also captures the behavioral effect of interest. It is not obvious what additional machinery the "Surprise" model offers, especially given that there is only one qualitative kind of surprise in the mid-bandit. Some simulation work might help elucidate how this model's predictions are different from the "Valence" model.

- I assume that the models were fit in the same way to all the data, such that surprises in the high/low bandits are treated the same as surprises in the mid bandit, and the same u/l/s parameters apply to all three. Is it possible that some scaling issues might be at play here? This could be tested by separating u/l/s by whether the distribution was bimodal or not.

- The main hypotheses are not explicitly stated - e.g. the introduction expanded enough on specific hypotheses regarding age and surprise-based learning. There are two parallel goals in this study: (1) impact of surprise on RL and (2) the impact of age on this relationship. The rationale and implications of these related aims is not fully developed.

- In the key behavioral analysis comparing choice of mid vs. low bandit, did the authors always condition on whether the previous trial's outcome was from the dominant mode? Otherwise it is possible to have several "surprising" trials in a row, which might induce repetition effects, potentially explaining the lack of a result in the modeling (as consecutive surprise trials are treated the same).

- Were participants aware of the distribution structure? How was the feedback schedule conveyed to them? The language used for instruction may impact participants’ overall surprise and uncertainty.

- The authors could do a better job at clarifying how de-correlated these surprise and uncertainty are in the "Surprise" model, which is like the "Uncertainty" model with a learning rate of 1.

Minor comments:

- The rationale for “free” and “guided” choices in the behavioral paradigm’s explanation is not clear.

- Including the equation for the LME models implemented would improve readability.

- Including some model equations in the illustrative Fig. 2 would also help with readability and working memory demands for interpreting betas

- Figure 2: k/l notation is confusing, why not say Left and Right?

- Line 252 – suggesting age-related neural differences is abrupt.

- AIC was used for model fitting – wouldn’t BIC be a better choice when comparing models with variable numbers of free parameters as this metric has a penalty for # of parameters.

- Figure 3c and e – difficult to tell which group-level comparisons are significant.

- Figure 3d – two older adults who displayed higher error for mid-high versus low-mid are unexpected. Does behavior correlate with estimate accuracy? How accurately do these participants estimate the likely outcome of respective choices? Are there individual differences in learning rates fit using the valence model related to performance (i.e., for these individuals I’d expect increased sensitivity to negative outcomes (higher α negative) which may explain the performance differences).

- The statement line 363 "the result cannot be explained by age group differences in risk aversion ..." was not clear

- Line 393 – typo “heighted learning from to such singular events”

- For the next sentence, are the authors referring to trials in which outcomes fall in the second mode of the mid distribution, but below the mean?

- Line 334 - are authors referring to the outliers in the OA or YA group? This seemed a bit arbitrary.

Reviewer #2: Attachment uploaded

**Have the authors made all data and (if applicable) computational code underlying the findings in their manuscript fully available?**

Reviewer #1: Yes

Reviewer #2: Yes

PLOS authors have the option to publish the peer review history of their article (what does this mean?). If published, this will include your full peer review and any attached files.

Reviewer #1: No

Reviewer #2: **Yes: **Jessica Schaaf, PhD
---

## [Decision Letter · Decision Letter 1]

16 Jul 2024

Dear Schuck,

We are pleased to inform you that your manuscript 'Influence of surprise on reinforcement learning in younger and older adults' has been provisionally accepted for publication in PLOS Computational Biology.

Please also note one of the reviewer's comment on the code repository. We would strongly encourage you to improve this alongsite the copyediting, so that the code can be executed once the paper will be published.

Best regards,

Tobias U Hauser, PhD

Academic Editor

PLOS Computational Biology

Thomas Serre

Section Editor

PLOS Computational Biology

Reviewer's Responses to Questions

**Comments to the Authors:**

Reviewer #1: Thank you for addressing the main issues I raised in my review! I found the posterior predictive check and additional modeling work and discussion compelling. The clarity of the manuscript is also improved. Overall I find this work interesting, thorough and an important addition to the literature.

Reviewer #2: Thank you for your clear and concise responses to my comments. I believe your changes have really improved the manuscript by clarifying the goal and by explicating your reasoning. I am a bit concerned about the model recovery as it seems that mainly Uncertainty effects can be recovered (Figure R5) and that the Surprise model and the Valence model can account for the same data (Figure R6). Yet, this should not preclude publication because all information is present for readers to evaluate your results themselves and because the authors nuanced their conclusions and acknowledge this shortcoming in the discussion. I therefore advise to accept your manuscript as is. Yet, I do have a suggestion for revising your open code base.

I tried running your code (specifically demographics.Rmd and results_01.Rmd) but did not manage to get it working. I think it is because the code contains self-written functions (I already ran into problems all the way in the beginning because Add_comp() and Get_plot_guides() did not exist) which resulted in me not being able to load the data and/or run the analyses. If I would want to fit your models, it would also greatly help me to have a README file pointing me to the necessary files and steps.

**Have the authors made all data and (if applicable) computational code underlying the findings in their manuscript fully available?**

Reviewer #1: Yes

Reviewer #2: Yes

PLOS authors have the option to publish the peer review history of their article (what does this mean?). If published, this will include your full peer review and any attached files.

Reviewer #1: No

Reviewer #2: **Yes: **Jessica Schaaf

---

## [Editor Report · Acceptance letter]

9 Aug 2024

PCOMPBIOL-D-23-01925R1 

Influence of surprise on reinforcement learning in younger and older adults

Dear Dr Schuck,

I am pleased to inform you that your manuscript has been formally accepted for publication in PLOS Computational Biology. Your manuscript is now with our production department and you will be notified of the publication date in due course.

With kind regards,

Anita Estes
